# A canonical neural mechanism for behavioral variability

Ran Darshan[1,2], William E. Wood[2], Susan Peters[3], Arthur Leblois[2] & David Hansel[2]

The ability to generate variable movements is essential for learning and adjusting complex behaviours. This variability has been linked to the temporal irregularity of neuronal activity in the central nervous system. However, how neuronal irregularity actually translates into behavioural variability is unclear. Here we combine modelling, electrophysiological and behavioural studies to address this issue. We demonstrate that a model circuit comprising topographically organized and strongly recurrent neural networks can autonomously generate irregular motor behaviours. Simultaneous recordings of neurons in singing finches reveal that neural correlations increase across the circuit driving song variability, in agreement with the model predictions. Analysing behavioural data, we find remarkable similarities in the babbling statistics of 5–6-month-old human infants and juveniles from three songbird species and show that our model naturally accounts for these 'universal' statistics.

[1] ELSC, The Hebrew University of Jerusalem, Israel, Edmond Jacob Safra Campus, Givat Ram 91904, Jerusalem. [2] Center for Neurophysics, Physiology and Pathology, Cerebral Dynamics, Learning and Memory Lab, CNRS-UMR8119 and University Paris Descartes, 45 Rue des Saints Pères, Paris 75270, France. [3] Trinity College of Arts and Sciences, Duke University, Durham, North Carolina 27708, USA. Correspondence and requests for materials should be addressed to D.H. (email: david.hansel@parisdescartes.fr).

Behavioural variability is a pivotal component of motor learning and adaptation[1,2]. While young individuals can usually produce non-stereotyped disorganized behaviours, motor exploration is more often expressed as movement variability around a stereotyped motor pattern. Highly irregular patterns of activity, which are ubiquitous in the brain[3], are thought to underlie variable motor behaviours[4,5]. Specifically in songbirds, a neural circuit necessary for song learning in juveniles[6,7] has been recently shown to be responsible for vocal variability both in adults[8] and throughout development[7,9,10]. This circuit includes two cortical-like areas: a premotor nucleus, the lateral magnocellular nucleus of the anterior nidopallium (LMAN), and its efferent motor nucleus, the robust nucleus of the arcopalium (RA). While RA is essential in driving the effectors (muscles or muscle synergies) producing the song[11], LMAN is not necessary for song production in adults[6,7] but has a key role throughout development and in adults in driving variability in the song[9,10] and in the activity of RA neurons[12].

The idea that temporally irregular activity of neurons in the central nervous system (CNS) is capable of generating behavioural variability may seem obvious. A careful examination, however, reveals that the link between irregularity in neural activity and behavioural variability is far from being straightforward. This is because to impact the behaviour, patterns of activity generated in the CNS must also be spatially correlated (that is, correlated across neurons). For example, consider the minimal model of a cortical network driving motor behaviour depicted in Fig. 1a. It consists of many neurons randomly connected recurrently, divided into $D$ functional groups; each group is composed of $M$ neurons (larger than $D$ by over an order of magnitude) that project to one effector of the motor behaviour. The collective dynamics of the network give rise to highly irregular firing patterns as a consequence of the interplay between excitation and inhibition[13] (Fig. 1b, Supplementary Fig. 1a,b). Despite this large variability in their activity, unless the number of neurons in a group is very small, fluctuations in the effectors are negligible (the coefficient of variation of the input to the effector, $CV_{eff}$, see Methods section, is very small; Fig. 1a, Supplementary Fig. 1e). This stems from the fact that the network activity is only weakly correlated across neurons (on the order of $1/N$, where $N$ is the number of neurons in the network, Fig. 1b right, Supplementary Fig. 1d) and thus, by virtue of the law of large numbers, the fluctuations they induce in the net input to an effector 'average out'. This example emphasizes the fact that, for the fluctuations to be transferred robustly from the CNS to the effectors, neuronal firing in the motor network must be sufficiently correlated within a neural population projecting to the same effector.

While the mechanism underlying asynchronous irregular spiking activity in recurrent networks of excitatory and inhibitory neurons is well understood[13–16], how the CNS autonomously generates patterns of activity, which are both temporally irregular and correlated across neurons, remains an open fundamental question[16–19]. A key result of our theoretical work is that the activity of neurons in the motor network driving the effectors will be highly irregular and also spatially correlated if this network receives topographically organized excitatory projections from another upstream strongly recurrent network, hereafter premotor network. In the context of the circuit driving song variability in songbirds, our theory predicts that correlations emerge along the LMAN–RA circuit, namely, that correlations across neurons are very weak in LMAN but substantial in RA. We validate this prediction with simultaneous extracellular recordings of neurons in singing finches. Our theory also suggests that vocal variability in different species of juvenile vocal learners should exhibit very similar statistics, as a consequence of universal statistical properties of the circuit dynamics. We verify this prediction by comparing the statistics of the song produced during the babbling phase of three species of songbirds as well as of human infants. Preliminary report of this work previously appeared in an abstract form[20].

## Results

We first show that temporally irregular and spatially correlated patterns of spiking activity can robustly emerge in a circuit of topographically organized and strongly recurrent networks. To this end, we consider the circuit depicted in Fig. 2a. Neurons in the motor network that project to the same effector share a fraction, $f$, of their premotor inputs, and this shared component is different from one group to the other (see Methods section for a detailed description of the architecture). With this architecture, the spiking activities of the neurons in the premotor, as well as in the motor network, are highly irregular, as a result of their recurrent dynamics. There is, however, an important difference between the networks in the spatial structure of their activities. In the premotor network, correlations across neurons are typically extremely weak (Figs 2b,c and 4a). In contrast, in the motor network pairs of neurons projecting to the same effector are substantially and positively correlated, whereas correlations are weak (and possibly negative) for neurons projecting to different effectors (Figs 2d–f and 4b and Supplementary Figs 3–5). These functional correlations are highly robust and only weakly influenced by the model parameters (Supplementary Figs 3d,e and 5d–f and Supplementary Note 1). As a result, fluctuations are amplified along the circuit (Fig. 2g) and the variability is robustly transferred to the effectors.

Importantly, the correlations in the motor network are substantial only if the footprint of the recurrent interactions in that network is sufficiently wider than the footprint of the premotor-to-motor projections (Fig. 2h). Indeed, when the recurrent interactions are too local, correlations in the motor network are weak (Fig. 2h,i). Thus temporally irregular and spatially correlated patterns of activity naturally emerge from the interplay between topographic feedforward (FF) projections from the premotor to the motor network and recurrent interactions within the motor network (Supplementary Fig. 10).

The emergence of spatial correlations in the motor network can be intuitively understood as follows. The FF input to a neuron in the motor network consists of one structured component, shared by all neurons belonging to the same functional group, and another one which is unstructured. Since the neurons in the premotor network are firing asynchronously, both components are the sum of a large number of uncorrelated contributions (on average $fK$ and $(1-f)K$, respectively; $K$ being the average number of synapses per neuron; see Supplementary Fig. 3i) and thus their temporal fluctuations are smaller than their temporal average by a factor on the order of $1/\sqrt{K}$ (Supplementary Fig. 3f: blue curve). Neural activity in the motor network will be spatially correlated if the amplitude of the fluctuations in the structured component to the network is on the order of the neuronal threshold. This implies that the temporally averaged FF input must be on the order of $\sqrt{K}$. This will happen if the strength of the FF connections are on the order of $1/\sqrt{K}$. To prevent the neurons in the motor network to fire regularly at a very high rate, the inhibitory recurrent inputs in the motor network must compensate for most of this averaged FF input. Such a compensation occurs naturally if the motor network is strongly recurrent and operates in the 'balanced excitation–inhibition' regime[13] (see Supplementary Note 1 for more details on the mechanism).

The fluctuations in the component common to all neurons in the same functional group give rise to the correlations in the

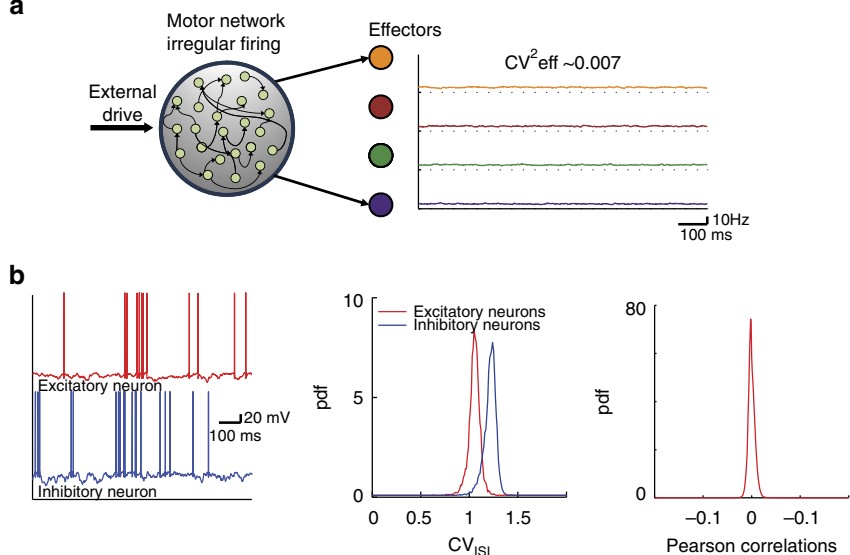

**Figure 1 | Fluctuations in the inputs to the effectors are very weak when noise is generated autonomously in the motor network.** (**a**) The motor network projects in a topographic manner to $D$ effectors ($D = 10$ effectors, 4 represented): each effector receives inputs from a different group of $M = 1,000$ neurons. In spite of the large variability of the neuronal activity, the variability of the effectors (right) is extremely small (coefficient of variation of the effector averaged over the 10 effectors: $CV^2_{eff} = 0.007$). (**b**) The neuronal activity in the motor network is highly irregular and the correlations across neurons are tightly distributed around zero. Left: Voltage traces for one excitatory (E, red) and one inhibitory (I, blue) neuron. Middle: Distributions of coefficient of variation of the inter-spike interval, $CV_{ISI}$. Right: probability density function (pdf) of Pearson correlation coefficients in the network.

activity in the motor network on a spatial scale on the order of the size of a group. Moreover, the groups also compete with each other provided that the recurrent interactions extend over a distance larger than the size of a group. As a result, the network dynamics self-organize such that the average instantaneous rates of the excitatory and inhibitory populations are essentially constant in time (Fig. 2d). This guarantees that the network operates in the balanced excitation–inhibition regime in a robust manner (see Supplementary Note 1).

**Correlation structure in circuit driving vocal variability.** Songbirds, with their well-identified and segregated circuit devoted to song learning, including a minimal circuit driving song variability (see Introduction), offer an ideal opportunity to test predictions of our theory. In songbirds, LMAN controls the trial-to-trial fluctuations across repetitions of the temporally structured song[8,9]. These fluctuations are important for adapting the song upon perturbations[21–23]. Moreover, anatomical studies in the circuit driving the song indicate that the projections from LMAN to RA are topographically organized, as our model posits for the projections in the premotor-to-motor pathway[24,25]. We therefore hypothesized that song variability stems from essentially uncorrelated fluctuations produced in LMAN, which by virtue of the topographic projections from LMAN to RA induce spatially correlated fluctuations in RA activity. To further test this hypothesis, we extended the two-area circuit considered above to take also into account the temporally structured inputs from nucleus HVC (used as a proper name) into RA neurons[12,26]. To this end, we included an additional FF excitation to the motor network in our model, representing the latter input (Fig. 3b, see Methods section). The responses of the neurons in the motor network are then locked to this input in a way that is reminiscent to the locking of RA neurons to the song[27,28] (compare Fig. 3a with Fig. 3c). However, these responses still exhibit trial-to-trial variability. By analysing the spatiotemporal patterns of these trial-to-trial fluctuations, we found substantial noise correlations

(see Methods section) for neurons in the motor network belonging to the same group but almost none in the upstream premotor network (Fig. 4). In the motor network, noise correlations were positive for pairs in the same group. They were typically weaker and negative for pairs in different groups. The averaged correlation over all pairs of excitatory neurons was very small due to the compensation between positive and negative correlations. (Fig. 4b and also Supplementary Fig. 4). Our model thus predicts a build-up of noise correlations along the circuit generating behavioural variability in singing birds.

To test this prediction, we recorded pairs of LMAN or RA neurons during singing in zebra finches. In LMAN, we found that spike-triggered average (STA) of the local field potential (LFP), as well as STA of the multi-unit activity were weak (Fig. 5b, Supplementary Fig. 6, see Methods section). We also found that noise crosscorrelograms were flat (Fig. 5c,d) and that correlation coefficients were tightly distributed around zero (Fig. 5e) in LMAN. By contrast, in RA neurons displayed substantial noise correlations during singing, as revealed by the shape of their crosscorrelograms (Fig. 5h,i; one-tailed two-sample $t$-test; $P < 0.01$ for single-units pairs, $n = 4$ pairs in LMAN and $n = 5$ pairs in RA; $P < 0.001$ for single- versus multi-units pairs, $n = 6$ pairs in LMAN and $n = 25$ pairs in RA; and $P < 0.001$ for multi-units pairs, $n = 21$ pairs in LMAN and $n = 21$ pairs in RA; see Methods section) and large values of noise correlation coefficients (compare Fig. 5j with Fig. 5e). The fact that correlations in RA were stronger than in LMAN is consistent with the model prediction, since the recorded units were likely to be located in the same functional group given the small distance between electrodes compared to RA diameter (see also Supplementary Fig. 6 and Supplementary Note 2). Multi-unit-STA and LFP-STA also consistently displayed high noise-related activity around the recorded spikes in RA, in contrast to LMAN (compare Fig. 5g with Fig. 5b and Supplementary Fig. 6a). Finally, we found that noise crosscorrelations (CCs) between LFPs recorded from evenly spaced electrodes decreased with the distance between the electrodes and became negative when they

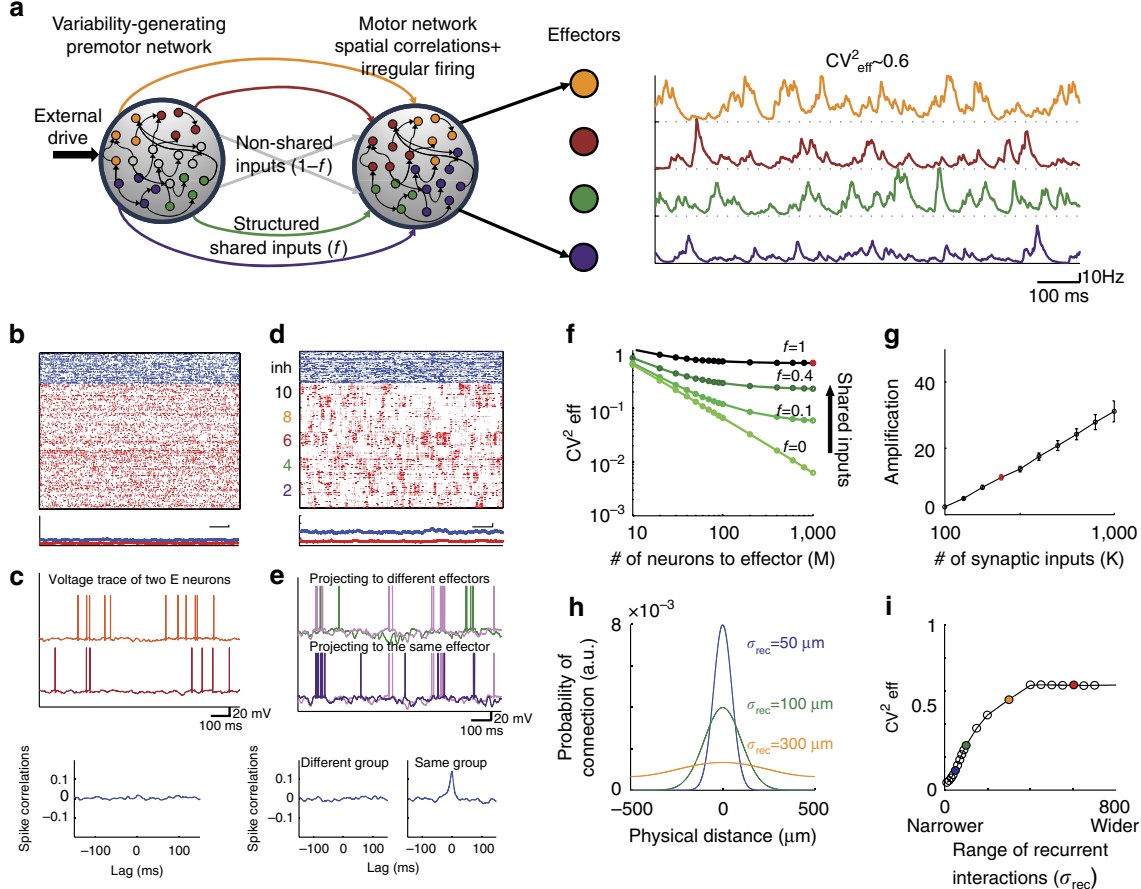

**Figure 2 | A generic neural circuit driving behavioural variability.** (**a**) When the premotor-to-motor projections are topographically organized, fluctuations in the inputs to the effectors are large. Left: circuit architecture. In the motor network, neurons in the same group (same colour, projecting to the same effector) share a fraction $f$ of premotor inputs (arrows coloured as the corresponding group) and have a fraction $1-f$ of non-shared inputs (grey arrows). Right: inputs to the effectors are variable $\left(CV^2_{eff} = 0.6\right)$. (**b,c**) In the premotor network, single neuron activity is irregular and neurons are very weakly correlated. (**b**) Top: Raster plots of E (red) and I (blue) premotor neurons. Bottom: instantaneous mean activity of E and I neurons (bars: 100 ms and 10 Hz). (**c**) Voltage of two excitatory premotor neurons (top) and their spike CCs (bottom). (**d,e**) In the motor network, single neuron activity is irregular and neurons are correlated. (**d**) Top: Raster plots of E and I populations. Bottom: instantaneous mean activity of E and I neurons (scale bar: 100 ms and 10 Hz). (**e**) Voltage traces of two neurons in the motor network projecting to different (top) and same (bottom) effectors. Bottom: pairs of neurons projecting to the same effector are substantially correlated (right); pairs projecting to different effectors are weakly correlated (left; see also Fig. 4). (**f**) The variability of the inputs to the effectors increases with the fraction of shared inputs and is substantial even if the number of inputs per effector, $M$, is large. This is because in the motor network the activities of the neurons belonging to the same group are correlated. (**g**) The circuit amplifies fluctuations. The amplification factor, $\frac{CV^2_{eff}}{CV^2_{inp}}$, (see Methods section) measures the ratio between the variability of the effectors ($CV^2_{eff}$) and of the input to the motor network ($CV^2_{inp}$). It increases linearly with the average number of synapses per neuron, $K$ (mean ± s.e.m.; see also Supplementary Fig. 3f). (**h**) Connection probability of two neurons in the motor network depends on their distance (see Methods section) with a footprint $\sigma_{rec}$. The diameter of the motor network is $\lambda = 1,000\ \mu m$. (**i**) $CV^2_{eff}$ decreases when narrowing the footprint of the recurrent interactions in the motor network. Red dot in the figure corresponds to the parameters used in **a**–**e**.

were far apart (Supplementary Fig. 6b and Supplementary Note 2). Therefore, as predicted by our model, noise correlations during singing are strong in RA, while they are extremely weak in LMAN.

Our electrophysiological recordings also reveal that the decays of the autocorrelations (ACs) and of the CCs of the spiking activity last for hundred of milliseconds in RA neurons (Figs 5h,i and 6b,c) and that these decays are substantially faster in LMAN (Fig. 6a,c; two-sample $t$-test, $P < 0.01$ for $n = 10$ single units in LMAN and $n = 14$ single units in RA). What is the source of the relatively slow decorrelations in the activity of RA neurons? In our theory, synchronous temporal fluctuations in RA activity will be slow if the shared FF drive of the neurons in the motor network slowly fluctuates (see Supplementary Note 1 and

Supplementary Fig. 7). If the synaptic dynamics in the premotor-to-motor pathway are slow, they give rise to autocorrelograms and crosscorrelograms in the motor network that can be as broad as in the data (compare Fig. 6a–c with Fig. 6d–f and Fig. 5h,i with Fig. 4b). This result suggests that the observed slowness of the fluctuations in RA activity stems from a low pass filtering of the fast fluctuations of LMAN outputs due to the large proportion of NMDA (N-methyl-D-aspartate) receptors in the LMAN-to-RA projections[29–31].

**Statistics of vocal variability in juvenile learners.** Juvenile songbirds produce babbling-like vocalizations that are not stereotyped and highly variable[10,32]. At this early developmental

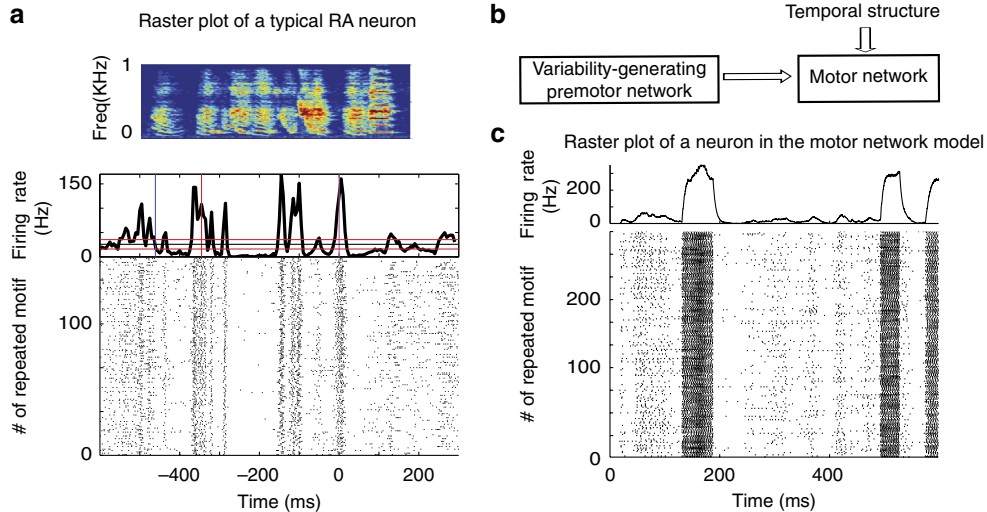

**Figure 3 | Single unit recordings in zebra finch RA nucleus and in the model motor network. (a)** Top: song motif of a zebra finch. Bottom: recordings of RA single unit over 133 repetitions of song motif, aligned to one syllable in the motif (lower panel) and the corresponding average firing rate (upper panel; 5 ms bin size). **(b)** Extension of the model depicted in Fig. 2a. Neurons in the motor network receives also temporally structured FF inputs, representing HVC inputs in the adult zebra finch (see main text and Methods section). **(c)** Raster plot and corresponding average firing rate of a neuron in the motor network of the model circuit in the presence of temporally structured FF input.

stage, the inputs from HVC to RA are not yet functional[33] and the song is mostly driven by LMAN–RA circuit[10,12]. We therefore asked whether the neuronal circuit depicted in Fig. 2a can drive behavioural variability with statistics similar to those observed during the babbling stage of juvenile birds. To this end, we combined the circuit with a mechanical model of the vocal production organ[34]. The latter was developed for zebra finches. To characterize the statistics of the output signal of the model, we computed the distribution of gesture durations (vocal elements) and the autocovariance of the envelope signal (ACE; see Methods section), which quantifies high-order correlations between consecutive gestures and inter-gesture intervals. The gestures durations had an exponential distribution in the model (Fig. 7a, bottom left; see Methods section). As for the ACE (Fig. 7a, bottom right), it monotonically decays over a duration of several tens to hundreds of milliseconds. Quantitatively, the decay time constant of the ACE and the scale parameter of gesture duration distributions depend on the synaptic time constant of the premotor-to-motor projections (compare the blue and red lines in Fig. 7a).

To what extent do these statistics depend on the details of the model architecture and connectivity, the neuronal dynamics and the nonlinearities in the input–output transduction by the effectors? Fluctuations in the FF input to the neurons in the motor network in our circuit consist of many uncorrelated fluctuating contributions and their statistics are thus close to Gaussian (Supplementary Fig. 8a). This is the case also for the net (FF + recurrent) input to these neurons. (Supplementary Fig. 8b). Hence, the fluctuating activity of the neurons in the motor network can be approximately described as a wideband Gaussian process that is rectified (Supplementary Fig. 8b–d), resulting in the tendency of the neurons to fire bursts of spikes with an approximately exponential distribution of durations (Supplementary Fig. 8f). Moreover, because neurons in the motor network are correlated, the temporal statistics of the input to an effector and of the neuron activity are similar. While the babbling behaviour generated by the circuit is a complex transformation[35] of the inputs to the effectors, the ACE or the distribution of gesture durations (but not finer structures of the gestures) are expected to be qualitatively independent on the details of this

transformation. For example, for rectified power-law transformations the distribution of gesture durations is close to exponential (Supplementary Fig. 8g) and the ACE barely depends on the nonlinearity (Supplementary Fig. 8e). This is also true when combining the circuit with a mechanical model of the vocal production organ[34]. Therefore, these features reflect universal statistical properties of the circuit dynamics and are, to a large extent, insensitive to the circuit parameters and to the details of the transformation from the input to the effectors to the vocal behaviour.

Given the universality of the statistics of the features analysed above in our model, we then compared the 'babbling behaviour' generated by the model with babbling behaviour in different vocal learners. To this end, we analysed babbling vocalizations of juveniles from three different songbird species with completely different adult repertoires (zebra finches: single song of 3–8 syllables per individual; swamp sparrow: 2–5 stereotyped songs per individual gathering 5–10 syllable types; canaries: complex song sequences based on a repertoire of 20–40 syllables per individual), as well as vocalizations of 5–6-month-old human infants (adult repertoire: complex sentences based on 10–100 phonemes grouped in > 10,000 words).

Remarkably, we found that the statistics of the vocalizations produced during the early period of babbling (but not later in development, Supplementary Fig. 9a–c,f) had a large degree of similarity in the four species we analysed. In all four species, as in the model, the distribution of vocal gesture durations could be well fitted with a single exponential (Fig. 7b–e, left and insets; see Methods section, see also[10,36]). In addition, the ACE lacks a clear temporal structure in all babbling vocalizations. The scale parameter of the gesture duration distributions, as well as the decorrelation times of ACE (that is, the typical time constant of the ACE) varied across individuals and species from several tens to a few hundreds of milliseconds (Fig. 7b–e). However, as the distributions were close to exponential and variability within species was small (Fig. 7f,h), interspecies and intraspecies differences in gesture duration distributions became comparable after normalizing each individual distribution by its species-averaged scale parameter (Fig. 7g–i; two-sample t-test, $P = 0.92$: only 9% of the total variance among distributions was attributed

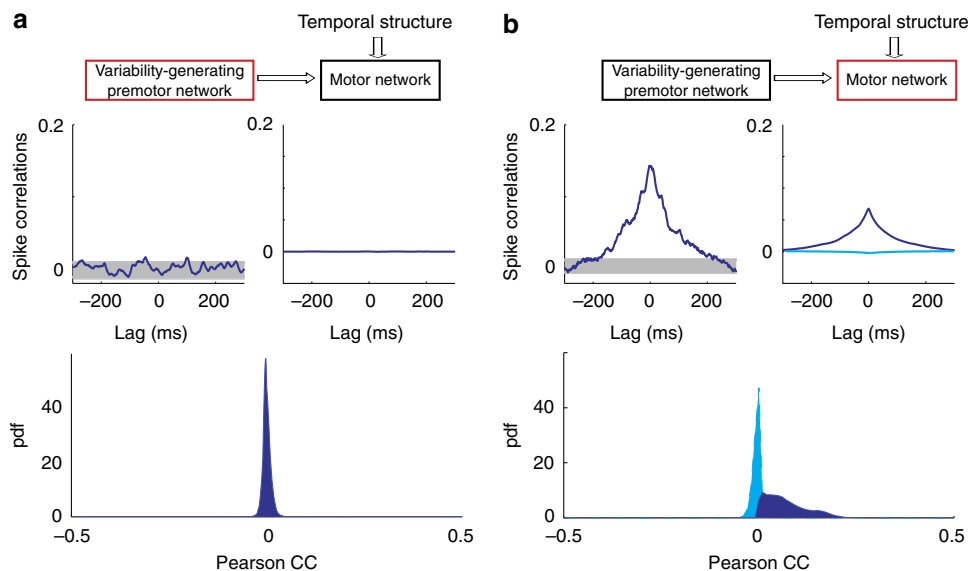

**Figure 4 | Correlations in the trial-to-trial neuronal variability increase along the model circuit generating motor variability.** (**a,b**) Noise correlations in the model. In the premotor network, noise CCs are weak. In the motor network, neurons activating the same effector have significant positive correlation coefficients. (**a**) Top left: example of noise CCs across two single units in the variability-generating premotor network (shaded area: 2.5 s.d. around the mean). Top right: population averaged CCs. Noise CCs are almost flat, indicating the absence of significant correlations in the activity of the premotor network. Bottom: Probability density function (pdf) of Pearson correlation coefficients in the premotor network. (**b**) Same as in (**a**) but for neurons in the motor network. Bottom: Conditional probabilities of the Pearson correlations of neurons in the same functional group (dark-blue; average correlations: ∼0.068) and neurons in different groups (light blue; average correlations: average correlations: ∼ − 0.0066). Note that the probability of having two neurons in a group and between groups depends on the number of groups, and by taking these priors into account, the average correlations across all neurons is close to zero (average correlations: ∼0.0008; see text and Supplementary Note 1).

to species differences compared to 87% before rescaling; one-way analysis of variance (ANOVA)). A similar uniformity was observed in the inter-gesture interval distributions after normalization by the species average (Supplementary Fig. 9d.; $P = 0.23$, only 10% of total variance among distributions was attributed to species differences compared to 66% before normalization; one-way ANOVA). Interspecies variations in ACE were mostly due to differences in the species-averaged decorrelation time ($P = 0.16$; only 17% of the total variance among ACEs was attributed to species differences, compared to 81% before normalization; one-way ANOVA). Finally, correlations between consecutive gestures and inter-gestures were small and comparable among species (Supplementary Fig. 9e). Together, these results show that in the four species we studied the statistics of the babbling-like vocalizations are very similar and can be naturally accounted by our minimal circuit.

## Discussion

Our paper addresses the extent to which the intrinsic temporal irregularity of neuronal activity in the CNS can drive motor variability. This is a fundamental non-trivial question since, as to impact the behaviour, patterns of activity generated in the CNS must also be spatially correlated (that is, correlated across neurons; see also Supplementary Note 3. for alternative mechanisms). Although the emergence of asynchronous irregular activity in recurrent networks is well understood[13,16], much less is known regarding the possible mechanisms giving rise to irregular spiking in which fluctuations of the activity are both temporally irregular and correlated across neurons. As a matter of fact, in virtually all network models of irregular spiking previously investigated, the activity is either asynchronous[16] or the synchronous component of the temporal fluctuations in neuronal spiking is strongly rhythmic[37].

In particular, previous theoretical studies[16,38] concluded that correlations should be very weak in strongly recurrent cortical circuits (on the order of $1/N$, where $N$ is the network size). However, these studies assumed a completely random connectivity, without structure (with an Erdös–Renyi graph). Here we showed that substantial correlations emerge naturally in a circuit with topography. With such an architecture, the dynamics self-organize in groups of neurons that are positively correlated within a group but negatively correlated between groups. In this spatial pattern of correlations, the balance between excitation and inhibition is maintained over the whole network. As a result, the circuit can eventually produce robust variable behaviour with 'universal' statistics. In fact, we showed that this mechanism does not require any fine-tuning of parameters. In particular, it is robust to the number of neurons and the average number of connections, as well as to the connectivity in the topographic pathway to the effectors (and the number of neurons projecting to an effector; see also Supplementary Note 1).

In songbirds, the organization of the LMAN-to-RA pathway becomes clearly topographic during the early sensory period of song learning[32]. Thus it is already present when juvenile finches start to babble (35–40 days post hatch (DPH)). Neurons in RA also send topographic projections to the hypoglossal nucleus (nXII) as well as to the respiratory motor nuclei[25]. The projections of the hypoglossal nucleus to syringeal muscles are also topographic[39]. Thus the pathway from RA to syringeal muscles (and likely similarly to respiratory muscles) is topographic, as required by our mechanism. Applied to the LMAN–RA circuit, this mechanism predicts that noise correlations are weak in LMAN but substantial in RA. We reported experimental evidence in line with this prediction in the adult zebra finch.

In juvenile and adult zebra finches, the inputs from LMAN to RA are dominated by NMDA receptors with slow kinetics of time

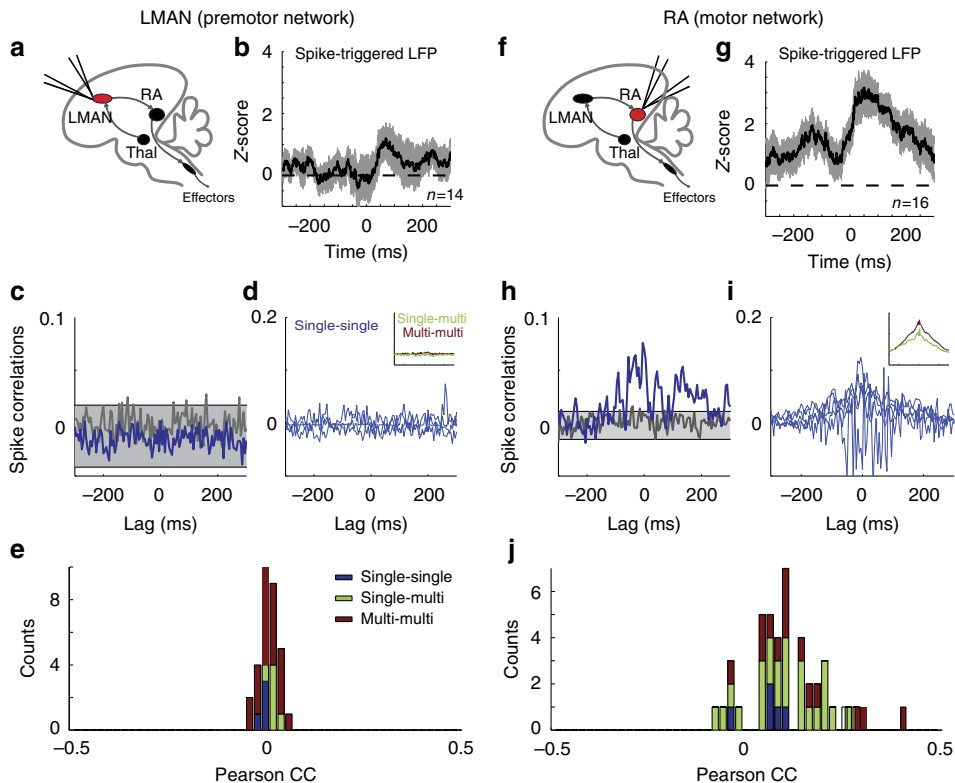

**Figure 5 | Correlations in the trial-to-trial neuronal variability increase along the circuit generating motor variability in singing birds.** Experimental recordings in zebra finches during singing. Noise correlations are weak in LMAN but substantial in RA. (**a**) Area of recordings. (**b**) STA of the noise LFP during singing in LMAN (mean ± s.e.m.). The motif-average LFP was subtracted from the LFP signal and the STA of this residual LFP was then computed separately for each single unit recording (see Methods section). (**c**) Noise CCs of two single units recorded simultaneously in LMAN during singing. The mean motif-related activity was subtracted from the instantaneous firing rate during singing and correlation analysis was performed on the residual trial-to-trial fluctuating signal (noise correlations, see Methods section). Noise CCs are flat after a random permutation of the spikes (grey trace, shaded area: 2.5 s.d. around the mean). (**d**) Single-unit pairs crosscorrelograms (blue) and average crosscorrelograms (inset) of single- versus multi-unit pairs (green) and pairs of multi-units (red) recorded from different electrodes in LMAN. (**e**) Distribution of Pearson correlation coefficients in LMAN. (**f–j**) Same as (**a–e**), but for neurons recorded in RA. In contrast to LMAN, RA neurons exhibit significant pairwise correlations. Crosscorrelograms are broad and their integrals are significantly larger in RA than in LMAN (see Methods and Results sections for statistical tests), reflecting the slow co-fluctuations in the activity of the simultaneously recorded units.

constant on the order of ~100 ms[29–31]. Moreover, recurrent excitation in LMAN is largely dominated by NMDA receptors and the kinetics of these receptors is faster in adults than in young juveniles[40], with typical time constants of ~30 ms in adults and ~120 ms in juveniles[41]. Therefore, slow synapses in LMAN as well in LMAN to RA projections can underlie the relative slowness of the dynamics of the babbling behaviour we reported in juvenile finches (see also Supplementary Fig. 7 and Supplementary Note 1). In agreement with this view, localized mild cooling of LMAN in zebra finches results in an increase in the time constant of the exponential gesture distribution during babbling-like behaviour and in a longer tail in the distribution in older juveniles[36].

Our behavioural data show substantial differences in the timescale of the babbling behaviour between zebra finches, canaries, swamp sparrows and humans. Our model suggests that this may be due to differences in the kinetics of NMDA receptors in these species. Revealing a direct correlation between these differences and NMDA receptors kinetics requires data on the latter. To the best of our knowledge, there is no such data available for canaries, swamp sparrows or human infants. However, the range spanned by the babbling timescales in our behavioural data is compatible with the diversity of kinetics reported in NMDA receptors of different subunit composition[42,43].

In adult subjects, motor variability is expressed as fluctuations around a stereotyped motor pattern, which despite their relatively small amplitude, can contribute significantly to motor learning[2]. At early stage of development, young animals, as well as human infants, produce spontaneous exploratory gestures referred to as 'motor babbling' that do not rely on any stereotyped or goal-oriented movement and rather appear to express pure motor variability[10,44,45]. Such exploratory movements may allow the self-organization[46] and the adaptation of sensory-motor networks through correlation-based (Hebbian learning) and reinforcement learning mechanisms[1,47–49]. These mechanisms posit that synaptic neural correlates of exploratory behaviour must persist for tens of milliseconds in the learning circuit. Our work suggests that the wide presence of NMDA receptors in the LMAN-to-RA projections[30,31] is a key component in the emergence of such eligibility trace in the overall dynamics of the circuit that generates behavioural variability in birds.

To conclude, we showed that a circuit comprising strongly recurrent neural networks, which is organized in a topographic manner, is capable of driving variable motor behaviours. This mechanism relies on only a few architectural constraints and is thus likely to be a general operating principle by which the brain acquires motor skills and adapt behaviour in a changing environment.

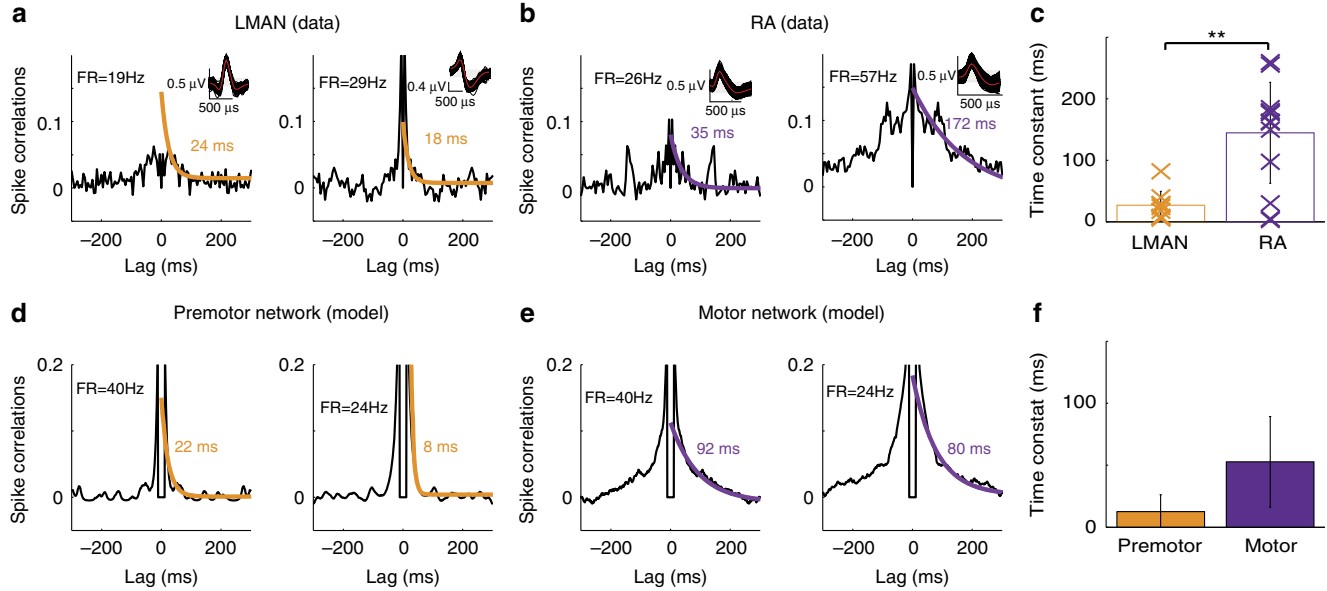

**Figure 6 | Temporal fluctuations slow down along the circuits generating motor variability (data + model).** (**a,b**) Decorrelation time in the activity of neurons in LMAN or RA during singing in zebra finches. (**a**) Two examples of noise ACs (see Methods section) for neurons recorded in LMAN (simultaneous recordings, CCs plotted in Fig. 5c). Inset: superimposed spike shapes (red: average trace). FR: average singing-related firing rate during song. (**b**) Same as (**a**) but for two RA neurons (simultaneous recordings, CCs plotted in Fig. 5h). The ACs are fitted to a decaying exponential (Orange in (**a**) and purple in (**b**); Time constant is indicated in the panels). (**c**) ACs are much broader in RA than in LMAN (single units and mean + s.d.). (**d–f**) The same as in (**a–c**) but in the model ($\tau_s^{E0} = 100$ ms, see Methods section). (**d**) ACs for neurons in the premotor network. (**e**) ACs for neurons in the motor network. (**f**) ACs in the premotor network decay faster than in the motor network (mean + s.d.).

During the review process, we became aware of a manuscript[50] which partially overlaps our work. It addresses the origin of spatially correlated activity in macaque V1. The mechanism proposed in this paper relies, similar to ours, on structured FF excitation and broad recurrent interactions.

## Methods

**Subjects.** Seven human infants (3 males and 4 females) were recorded in their natural environment. Their parents gave written informed consent for participation in this study. Nine zebra finches and five canaries were obtained from our breeding facilities (Paris Descartes and Paris Sud Universities). Seven swamp sparrows were collected as nestlings and hand-reared in the laboratory (see ref. 51 for details). Birds were housed under natural light/dark conditions and provided with food and water *ad libitum*. Animal care and experiments were performed in accordance with European directives (86/609/CEE and 2010-63-UE) and the French legislation. Experiments were approved by Paris Descartes University ethics committee.

**Human infants.** We recorded spontaneous vocalizations in six infants in their natural environment starting from 5 to 7 months after birth (denoted as the 'babbling period'). The parents were instructed to place a recorder (digital dictation machine with stereo microphone, ICD-PX333M SONY) near the baby's head for ~30 min at least 5 days a week for several weeks (4–20 weeks). The data presented include babies for which vocalizations were collected from at least 20 days during this recording period. Additionally, one 10-month-old infant was recorded for repetitive babbling.

**Zebra finches.** Juvenile zebra finches were raised in single cages with their parents and siblings. At age 26–41 DPH ('babbling period'), 9 male zebra finches were removed and placed in custom-made sound isolation chambers. Vocalizations were recorded for 10–30 days continuously with Sound Analysis[52], which was configured to ensure that recordings were triggered on all quiet vocalizations of young birds. Five of the nine birds were continuously recorded until song crystallization (~3 months), with episodic access to their father.

**Swamp sparrows.** Seven swamp sparrow males were recorded in individual sound isolation chambers (Industrial Acoustics AC-1) once per week, starting in February of their first year when they were about 250 DPH. The onset of song development was first detected at 262–296 DPH ('babbling period'), and recording continued up to 366–386 DPH, when the males were singing crystallized adult song. Subsong was sampled for 30 min (Marantz PMD221 cassette tape recorder, Realistic Omnidirectional microphone, Yamaha Mike to Line Amplifier). An automated system was introduced to detect and record song during late plastic and crystallized song using a voice-activated switch (modified UherAkusomat) and a Digital Delay System (Digitech).

**Canaries.** Juvenile canaries were raised in our breeding facility at Paris Sud University in single cages with their parents and siblings. At age 75–150 DPH ('babbling period'), as they started to produce their first vocalizations, five male canaries were removed and placed in custom-made sound isolation chambers. Vocalizations were recorded continuously for 3 months (September–December) during the fall following their birth with Sound Analysis Pro, which was configured to ensure that recordings were triggered on all quiet vocalizations of young birds. Four of the five birds were also recorded 3 months later (early spring) for 5–10 more days.

**Vocalizations.** Songs and infant vocalizations were manually sorted. For subsongs, we took the first recorded song vocalizations of the bird. Recordings were from 1 day of vocalizations, except for zebra finches, where in some individuals subsongs from 1 to 3 recording days were combined to get enough gestures.

**Spectrograms.** Spectrograms were estimated using the multitaper method with two slepian tapers.

**Envelope signal.** We extracted the envelope of the signal (termed also 'amplitude' in the literature) by band passing the sound signal in the frequency ranges of the vocalizations (from 800 Hz and up to 4,000–10,000 Hz, depending on the species, with order-80 linear-phase finite-duration impulse response filter), taking the absolute value of the signal and low passing it at 1–200 Hz with a linear filter of order-200 linear-phase filter finite-duration impulse response.

**Averaged ACs of envelope (ACE).** The ACE was estimated for each recording and then normalized to the zero lag. The ACE signal was then estimated by averaging this signal over 1 day of recording sessions.

**Gesture and inter-gesture segmentation.** We used a local method for gesture and inter-gesture detection. We calculated the peaks of the derivative of the log-envelope signal (after band passing the signal; see above) that was smoothed

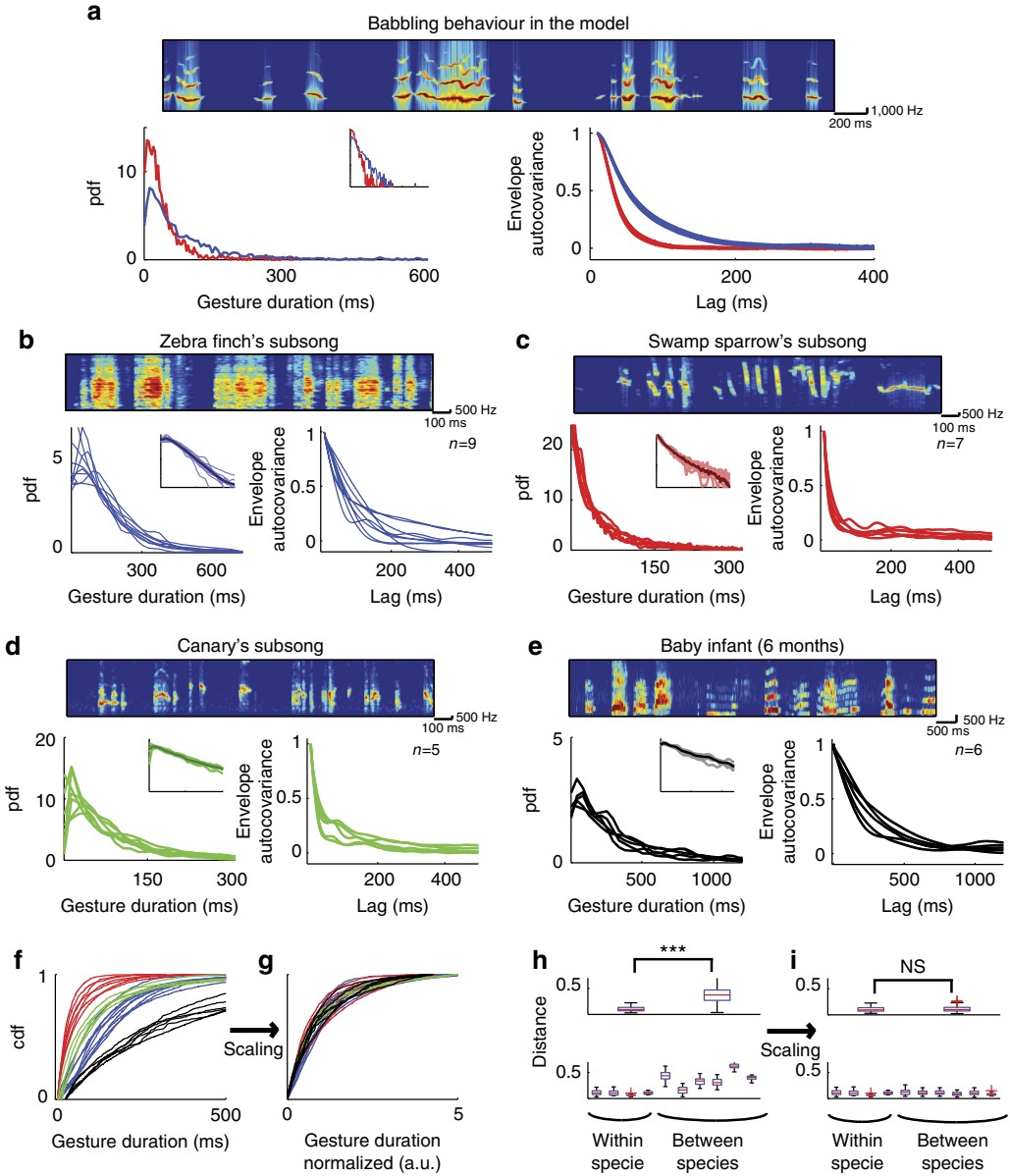

**Figure 7 | Babbling statistics are similar across different vocal learners and in the model.** (**a**) Statistics of the babbling behaviour generated by the model circuit depicted in Fig. 2a–e when coupled to a minimal model of the vocal organ (see Methods section). Top: spectrogram of the vocal output signal $\left(\tau_s^{EO} = 100 \text{ ms}\right)$. Bottom: probability density function (pdf) of vocal gesture durations (left) and averaged autocovariance of the envelope (ACE; right). Inset: distribution of gesture durations when the y axis is in log-scale. The distribution of gesture durations is well approximated by an exponential with a 'scale parameter', $\tau_{gesture}^{model}$ (see Methods section). ACE decorrelates over a time duration of $\tau_{ACE}^{model}$. Slow synaptic dynamics in the premotor-to-motor projections (red: $\tau_s^{EO} = 50 \text{ ms}$; blue: $\tau_s^{EO} = 100 \text{ ms}$) results in slowly fluctuating vocal output (red: $\tau_{gesture}^{model} = 60 \text{ ms}$ and $\tau_{ACE}^{model} = 31 \text{ ms}$; blue: $\tau_{gesture}^{model} = 120 \text{ ms}$ and $\tau_{ACE}^{model} = 64 \text{ ms}$). (**b–i**) Statistics of the babbling behaviour in four species of vocal learners (ages of the subjects ('babbling period') are given in Methods section). Blue: Zebra finches (Zf); Red: Swamp sparrows (Sw); Green: Canaries (Ca); Black: Human infants (Bab). Different lines of the same colour correspond to different subjects from the same species. (**b–e**) Same as in (**a**), but for the Zf (**b**: compare to the blue line in **a**), Sw (**c**: compare to the red line in **a**), Ca (**d**) and Bab (**e**). Gesture duration distributions lack any clear peak and are well fit with exponential decaying function with scale parameters (mean ± s.e.m.): $\tau_{gesture}^{Zf} = 138 \pm 8 \text{ ms}$; $\tau_{gesture}^{Sw} = 44 \pm 4 \text{ ms}$; $\tau_{gesture}^{Ca} = 108 \pm 4 \text{ ms}$; $\tau_{gesture}^{Bab} = 337 \pm 17 \text{ms}$. The ACE decay time is specie-dependent: $\tau_{ACE}^{Zf} = 80 \pm 7 \text{ ms}$; $\tau_{ACE}^{Sw} = 23 \pm 2 \text{ ms}$; $\tau_{ACE}^{Ca} = 42 \pm 7 \text{ ms}$; $\tau_{ACE}^{Bab} = 258 \pm 23 \text{ ms}$. (**f,g**) Cumulative distribution functions (cdf) of gesture duration for the four species before (**f**) and after (**g**) normalizing the gesture durations by $\tau_{gesture}^{species}$. (**h**) Top: Interspecies differences in cdfs are much smaller than intraspecies differences (Kolmogorov–Smirnov statistic as a distance measure between cdfs). Bottom: Differences of cdfs in pairs of learners within (left to right: Zf–Zf, Sw–Sw, Ca–Ca, Bab–Bab) and between species (left to right: Zf–Sw, Zf–Ca, Zf–Bab, Sw–Ca, Sw–Bab, Ca–Bab). (**i**) Most of the interspecies differences in (**h**) are accounted for by normalizing the gesture durations to the scale parameter of the exponential fit of their distributions (see Results and Methods sections for statistical comparisons).

by a 5–30 ms sliding window, depending on the noise level of the signal and the species (using fpeaks in Matlab and a filter of ($-$1 0 1)) and defined sound onsets and offsets as the closest points to these crossings. We defined the threshold for each file by the $x$ percentile of the peaks, where $x$ was in the range of 85–98. The percentile threshold, $x$, as well as other relevant parameters, for segmentation were fixed after manually examining a subset of the data for each recording day. When the signal was too noisy to use the local method (mainly for infant babbling and several swamp sparrows), we used a global threshold: for each recording, we calculated a sound threshold by fitting a two-Gaussian mixture model (corresponding to noise and sound) using an expectation-maximization algorithm for the log-envelope signal. We then detected crossings of this threshold and defined sound onsets and offsets as the closest points to these crossings where the envelope deviated from the noise by 4 s.d. Using a global method instead of a local one on all the juvenile recordings yielded similar results; however, we preferred to use the local method whenever possible. For both methods, sounds separated by a duration of $<$7 ms of silence were merged into a single gesture, and segments of overly long (Zf: $>$800 ms; Sw: $>$400 ms; Ca: $>$900 ms; Bab: $>$3,500 ms) or short durations (Zf, Sw, Ca: 7 ms; Bab: $<$30 ms) were eliminated.

**Fitting exponential decay.** We fit an exponential function to the gesture duration distribution using maximum-likelihood estimation on a finite interval[36] (based on a median of 1,870 gestures per day for a songbird and 700 for about 1 month of human infant recordings; interval duration: Zf: 50–800 ms; Sw:10–200 ms; Ca: 50–600 ms; Bab: 50–1,500 ms). Distributions that were well fit by the exponential function usually had a high goodness-of-fit metric[36] (adjusted $R^2 > 0.7$ and Lilliefores statistic $<$3.5). To extract decorrelation timescales from the ACE, we fit an exponential decay to the ACE.

**Microdrive implantation.** For single- and multi-unit recordings, a custom-built motorized microdrive (RP Metrix) was modified to accept 2–4 tungsten microelectrodes (8–20 M$\Omega$, FHC), as well as lateral positioner. It was implanted in LMAN or RA as follows: Young adult male zebra finches ($<$180 DPH, 3 for nucleus LMAN and 2 for nucleus RA) were anaesthetized with 5% isoflurane (induction) and placed in a stereotaxic apparatus with a head angle of 30–50° (for LMAN implantation) or $-$5° (for RA). Anaesthesia was maintained with 0.5–1% isoflurane for the duration of the surgery. LMAN/RA was located using previously established stereotaxic coordinates and identified based on its characteristic neural activity patterns. The electrodes and exposed brain were surrounded with Kwik-Cast (WPI), and the microdrive was secured to the skull using dental cement (Superbond, Phymep). A silver wire implanted under the skull acted as a ground, and a low impedance fixed tungsten electrode served as the reference. We also conducted LFP recordings in nucleus RA, using 3 × 3 microelectrode arrays (AlphaOmega) with an impedance of 1–2 MOhm. Electrode arrays were implanted in young male zebra finches under isoflurane anaesthesia (as specified above), relying on the recorded signals to locate nucleus RA, and then fixed onto the skull using dental cement (Superbond, Phymep). A silver wire implanted under the skull was used as a ground, and one of the contact points of the array served as the reference. In both types of experiments, subjects were allowed to recover and habituate to the weight of the recording apparatus for a few days. They were then transferred to the recording cage and connected through a commercial tether and head stage (Neuralynx or AlphaOmega) and the implanted microdrive to a mercury commutator on the roof of the cage (Dragonfly systems). An elastic thread built into the tether helped to support the weight of the implant. Subjects remained tethered both during and between experiments.

**Chronic recordings.** Neural signals and vocalizations were collected using a commercial head stage and acquisition system (Neuralynx or AlphaOmega). Signals were amplified, digitized and filtered either $<$300 Hz (LFP signal) or between 300 and 30 kHz (spike signal).

**Data analysis.** Spike signals were analysed using Spike2 software (Cambridge Electronic Design) and custom-written software in Matlab (MathWorks). Single- and multi-unit signals were isolated using Spike2, and spike times were then exported to Matlab. Motif onset times were extracted from sound recordings using custom programs. We calculated the autocorrelograms (AC; 5 s window, 5 ms bin) of single spike trains and crosscorrelograms (CC; 5 s window, 5 ms bin) of all pairs of spike trains recorded simultaneously. Activity was first aligned to motif onsets and then averaged over all motifs produced during each recording session (peristimulus time histogram (PSTH) analysis), using a time window limited to the duration of a single song motif, and 5 ms bins. To eliminate temporal variability due to fluctuations in the duration of single syllables, spike trains were aligned and stretched using piecewise linear time warping with each syllable onset as a time reference[28]. Signal correlations measure the similarity of the activity of two neurons during singing. Noise correlations, on the other hand, is a measure of the similarity of the trial-to-trial variability (around the motif-related PSTH) of two neurons. We computed the spike counts in 5 ms bins during song production and subtracted the mean motif-related PSTH of a song motif for each neuron for all motifs produced. For each pair of simultaneously recorded neurons (by definition, the noise correlations can only be calculated for simultaneously recorded neurons),

we computed the noise correlations by calculating the correlation coefficient of these two vectors. To compare the shapes of the crosscorrelograms from units recorded in two considered brain nuclei (LMAN and RA), we measured the mean deviation from zero in a crosscorrelogram according to the following procedure. The absolute value of the CC function was first averaged over the ($-$50 ms $+$50 ms) window (appropriate for the behavioural output given the integration time-constant). The absolute value of the AC functions of the two corresponding units was also averaged over the same time window. The average absolute CC was then normalized to the square root of the product of the average absolute AC functions to provide the numbers used in the statistical tests in the main text. This equation is given by:

$$\text{Shape of crosscorrelograms} = \frac{\sum_{\tau=-50\,\text{ms}}^{\tau=50\,\text{ms}} |c_{12}(\tau)|}{\sqrt{\sum_{\tau=-50\,\text{ms}}^{\tau=50\,\text{ms}} |c_{11}(\tau)| \sum_{\tau=-50\,\text{ms}}^{\tau=50\,\text{ms}} |c_{22}(\tau)|}}$$

where $c_{12}(\tau)$ is the crosscorrelogram across the two neurons at time lag $\tau$ and $c_{11}(\tau)$ and $c_{22}(\tau)$ are the ACs of the two neurons.

LFP signals were aligned to motif onsets and averaged over all motif renditions produced during a recording session. To calculate the 'noise STA LFP', we first subtracted motif-aligned average LFP from LFP signals recorded during each single motif. We then computed the average of the residual-LFP signals cut in 600 ms window around each spike over all spikes produced during all motif renditions in the recording session. Additionally, we computed noise STA of the envelope of the background multi-unit spiking activity present behind single-unit recordings. To this end, we first removed spike shapes from the sorted single unit from raw spiking signal, rectified the leftover background signal and convolved it with a 5 ms wide Gaussian function. A similar treatment as for the LFP was applied to this background multi-unit envelop to get its noise STA.

To quantify the possible effect of bad or partial time warping on the level of correlation in our data set, we first compared the level correlation in our data set before and after time warping and then also incorporated artificially wrong syllable timing before the time warping was applied. To this end, we added variable jitter (from 2 to 500 ms) to the syllable onset times and then time-warped the spike times of the two units according to this jittered syllable timing. This manipulation served to artificially introduce a strong misalignment of the spiking activity with real singing behaviour, with a common time jitter for both units, without modifying the average activity of the neurons in a given trial.

**Statistics.** Numerical values are given as mean $\pm$ s.d., unless stated otherwise. Whenever using a statistical test, we report the type of test applied and the associated $P$ value (probability of observing the given result, or one more extreme, by chance if the null hypothesis is true).

**Histology.** After the last recording session, subjects were killed by intramuscular injection of sodium pentobarbital (Nembutal) and perfused transcardially with 0.9% saline followed by 4% paraformaldehyde as fixative. The brain was then removed, postfixed in 4% paraformaldehyde for 24 h and cryoprotected in 30% sucrose. Sections (60 $\mu$m thick) were then cut in the parasagittal plane on a freezing microtome and processed for histological examination to verify the location of the recording electrodes. Tissue was Nissl stained to visualize the electrode tracks.

**Spiking network model.** Our model consists of two large recurrent networks, both comprising $N_E$ excitatory (E) and $N_I$ inhibitory (I) neurons. For simplicity, we take $N_E = N_I = N$. These two networks represent a premotor and a motor network. The premotor network projects in a FF manner to the motor network, and the latter activates a small number of effectors, consistent with the songbird anatomy[10,24,25,39].

**Single neuron and synaptic dynamics.** All the neurons in the circuit are modelled as leaky-integrate-and-fire (LIF) units. The subthreshold dynamics of the membrane potential, $V_{i,\alpha}(t)$, of neuron $i$ in population $\alpha(i=1, \ldots, N_\alpha; \alpha = \text{E,I})$ obey:

$$\tau_m \frac{dV_{i,\alpha}(t)}{dt} = -\left(V_{i,\alpha}(t) - V_L\right) + I_{rec,i}^\alpha(t) + I_{FF,i}^\alpha(t)$$

where $\tau_m$ is the neuron membrane time constant, $I_{rec,i}^\alpha(t)$ is the recurrent input into neuron ($i$, $\alpha$), due to its interactions with other neurons in the same network (premotor or motor), $I_{FF,i}^\alpha(t)$ is the total FF input into that neuron. $V_L$ is the reversal potential of the leak current (taken to be $V_L = -60$ mV).

These subthreshold dynamics are supplemented by a reset condition: if at $t = t_{i\alpha}$ the membrane potential of neuron ($i$, $\alpha$) crosses the threshold, $V_{i,\alpha}(t_{i\alpha}^-) = 10$ mV, the neuron fires an action potential and the voltage reset to $V_{i,\alpha}(t_{i\alpha}^+) = -60$ mV.

We model all synaptic inputs as pure currents. The total current into neuron ($i$, $\alpha$) due to its recurrent interactions yields:

$$I_{rec,i}^\alpha(t) = \sum J_{ij}^{\alpha\beta} S_j^{\alpha\beta}(t)$$

where $J_{ij}^{\alpha\beta}$ is the strength of the connection from presynaptic neuron ($j$, $\beta$) with

neuron $(i, \alpha)$ and $S_j^{\alpha\beta}(t)$ are the synaptic variables, which follow the dynamics:

$$\tau_s^{\alpha\beta} \frac{dS_i^{\alpha\beta}(t)}{dt} = -S_i^{\alpha\beta}(t) + \sum_{\{t_{j\beta}\}} \delta(t - t_{j\beta}).$$

Here $\tau_s^{\alpha\beta}$ is the synaptic time constant (assumed to depend solely on the nature—excitatory or inhibitory—of the presynaptic and postsynaptic neuron) and the sum is over all spikes emitted at times $t_{j\beta} < t$.

**Recurrent architecture.** The recurrent connectivity of the E and I populations in the premotor network is random (Erdös–Renyi graph). In each network, the connectivity matrix, $C^{\alpha\beta}$, between presynaptic population $\beta$ and postsynaptic population $\alpha$ is therefore a random $N \times N$ matrix such that $C_{ij}^{\alpha\beta} = 1$, with probability $K/N$ and zero otherwise, where $K$ is the average number of inputs a neuron receives from population $\beta$. We assume that the strength of the synapses depends solely on these populations yielding: $J_{ij}^{\alpha\beta} = J_{\alpha\beta} C_{ij}^{\alpha\beta}$ where $J_{\alpha E} > 0$ (excitation) and $J_{\alpha I} < 0$ (inhibition). When comparing the dynamics of networks with different connectivity, we follow the prescription[13,53]:

$$J_{\alpha\beta} = \bar{J}_{\alpha\beta} / \sqrt{K}$$

where the parameters $\bar{J}_{\alpha\beta}$ are of order unity and can be different for the premotor and motor networks.

**Distance-dependent recurrent architecture.** In the motor network, the connectivity is random with probability, which depends on the distance between the neurons. The probability of connections between two neurons is $p_{\alpha\beta}^{ij} = \frac{K}{N} f(x_{i\alpha} - x_{j\beta})$, where $x_{i\alpha} = i\frac{\lambda}{N}$ is the location of neuron $i = 1, ..., N$ in population $\alpha$, and

$$f(x_{i\alpha} - x_{j\beta}) = \frac{1}{\sqrt{2\pi\sigma_{rec}^2}} \sum_{m=-\infty}^{\infty} e^{-\frac{(x_{i\alpha} - x_{j\beta} + m)^2}{2\sigma_{rec}^2}}$$

where $\sigma_{rec}$ is the footprint of the recurrent interactions. Here we have assumed for simplicity that the motor network is one dimensional, of size $\lambda$ and periodic boundary conditions. For large values of $\sigma_{rec}$, distant neurons are as likely to be connected as close ones, while for small values of $\sigma_{rec}$ only neurons which are close have a significant probability to be connected (Fig. 2h). In most of the results depicted in the paper, we assume that the recurrent interactions in the motor network have a wide footprint ($\sigma_{rec} \to \infty$), except for Fig. 2h,i, where we investigate how the results depend on the value of $\sigma_{rec}$.

**Feed-forward architecture.** The premotor network receives external FF inputs, which in the context of the songbird system represent the thalamic (the medial part of the dorsolateral nucleus of the anterior thalamus, DLM) inputs that may tonically activate LMAN during song. The total number of FF inputs to a premotor neuron is modelled as a constant drive[13,53], $\sqrt{K} \bar{I}^\alpha$. Similarly, the motor network receives a FF input from outside the circuit that we model as a constant drive. Importantly, the motor network also receives FF projections from the premotor area that exhibit a topographic organization. To implement this key feature of the architecture of our model, we divide the excitatory population in the motor network into $D$ statistically equivalent functional groups ($N/D$ neurons in each group). For each group, we choose a set of $fK$ neurons (set $P_l$) in the premotor network projecting to in neurons in the group. For each neuron in the group, additional inputs are chosen by drawing randomly from the premotor network with probability $(1-f)K/N$. Each neuron in a group therefore receives on average $K$ projections from the premotor network. Changing $f$ allowed us to easily manipulate the total amount of correlations in the FF inputs by changing the parameter, $f$, keeping the total average number, $K$, of premotor inputs per neurons fixed. For $f = 1$ all the projections are topographic, whereas if $f = 0$ they are completely random. The total FF premotor input into an excitatory neuron $i$ ($i = 1, ..., N$) in group $l$ ($= 1, ..., D$) in the motor network is therefore modelled as:

$$I_{FF,il}^E(t) = J_{E0} \left\{ \sum_{j \in P_l} S_j^{E0}(t) + \sum_{j \in premotor} C_{ij}^{E0} S_j^{E0}(t) \right\} + \sqrt{K} \bar{I}^E$$

In this architecture, the probability that two neurons in group $l$ share premotor inputs is $f + O(K^2/N^2)$. Note that if $K$ is too large it will be impossible to have different shared inputs for each group in the motor network. However, this does not happen with the model parameters in the simulations described in the paper since we take $N \geq 10{,}000$, and the maximum number of connections is $K = 1{,}000$ for 10 clusters (Supplementary Fig. 3d).

The total FF premotor input into an inhibitory neuron $i$ ($i = 1, ..., N$) in the motor network is $I_{FF,i}^I(t) = J_{I0}\{\sum_{j \in premotor} C_{ij}^{I0} S_j^{I0}(t)\} + \sqrt{K} \bar{I}^I$. The synaptic strength, $J_{\alpha 0}$, is parameterized as the recurrent synapses: $J_{\alpha 0} = \bar{J}_{\alpha 0}/\sqrt{K}$, with $\bar{J}_{\alpha 0}$ of order unity and $C_{ij}^{\alpha 0} = 1$ with probability $K/N$ and zero otherwise. Finally, similar to the recurrent interactions, the dynamics of the synaptic variables $S_i^{\alpha 0}(t)$ yields: $\tau_s^{\alpha 0} \frac{dS_i^{\alpha 0}(t)}{dt} = -S_i^{\alpha 0}(t) + \sum_{\{t_{jE}\}} \delta(t - t_{jE})$ with

$t_{jE}$ as the spike times of neuron $(j, E)$ in the premotor network. For simplicity, we take $K_{FF} = K$.

**Temporally structured FF input to the motor network.** We model the temporally structured inputs to the motor network (HVC to RA input) by including an additional contribution, $I_i^{struct}(t)$, to the FF input received by the neurons in this network. The input, $I_i^{struct}(t)$, to neuron $i$, lasts for a duration of 600 ms (a typical duration of a zebra finch song motif) repeated 300 times. It consists of a random sequence of On and Off periods, the duration of which are drawn from an exponential distribution with mean 20 ms for the On and 70 ms for the Off periods. The amplitudes of the input during the On periods are drawn randomly from uniform distribution over an interval [0.1, 0.5]. The input sequences are generated independently for neurons in different functional groups. Each group is then divided into 20 (non-overlapping) subgroups such that all the neurons in a subgroup share the sequence. Note that the results depicted in Fig. 4a were obtained by simulating the network without structured input.

**Effectors.** The pathway from the motor network to the effectors is topographic. Specifically, we assume that: (1) the number of effectors and the number of groups are equal; (2) a given effector is activated by $M \gg D$ neurons in the motor network randomly chosen from one group; and (3) different effectors are activated by different functional groups. We modelled the activation of an effector, $E_l(t)(l = 1, ..., D)$, as a linearly filtered version of the activity of the neurons in the motor network, namely:

$$\tau_{eff} \frac{dE_l(t)}{dt} = -E_l(t) + \sum_{\{t_{jE}\}} \delta(t - t_{jE})$$

where $\tau_{eff}$ is the effector time constant and the sum is over all spikes emitted by the neurons in the motor network, which activates the $l$ effector at times $t_{jE} < t$.

**The vocal organ.** We modelled the vocal tract as in Amador et al.[34] In particular, we did not include the trachea or the Helmholtz filter, as these filters are species specific and in general will not affect gesture and inter-gesture durations. Two variables activate the vocal organ: tension and pressure. We modelled the pressure variable as $\widetilde{Pr} = [E_1]_+$, where $[x]_+$ is a rectified linear function. The tension is modelled as a linear combination of nine effectors: $\widetilde{T}_n = \frac{1}{D-1} \sum W_a E_a$, where, $W_b$ ($l = 2, ..., D$), are random weights, $W_l \sim N(0,1)$. Tension and pressure are then scaled to fit the dynamic range of the oscillating phase (see ref. 34):

$$T_n = \mu_T + \sigma_T z_T$$

and

$$Pr = \frac{[E_1 - E_1]_+}{\max([E_1 - E_1]_+)} P_{max} - b$$

where $z_T$ is the z-score of $\widetilde{T}_n$ and $\sigma_T$, $\mu_T$ and $P_{max}$ are constant parameters that define the dynamic range and $b$ is a bias that ensures that when there is no pressure the system is at a fixed point. We take: $\mu_T = 0.6$; $\sigma_T = 0.2$; $P_{max} = 0.21$; $b = 0.01$. The tension and pressure were then smoothed by a rectangular window of 20 ms and interpolated to a sampling frequency of 44,100 Hz. We then used the tension and pressure parameters to simulate the model by Amador et al.[34] Finally, to reduce transient effects at the boundaries of the gestures (as a result of crossing the bifurcation) generated by the vocal tract model, the sound signal was taken as the product of the output model and the Pr signal.

**Model parameters.** Unless specified otherwise, the parameters used in the simulations were: $N = 10{,}000$; $K = 400$; $D = 10$; $\tau_m = 10$ ms; $\tau_{eff} = 10$ ms. In the simulations depicted in Fig. 1, synaptic strengths and external FF inputs were: $\bar{J}_{EE} = 0.5$, $\bar{J}_{IE} = 3$, $\bar{J}_{EI} = -1.5$, $\bar{J}_{II} = -2$, $\bar{I}^E = 0.2$; $\bar{I}^I = 0.1$ for the premotor as well as for the motor network. All synaptic time constants were 3 ms and $\bar{J}_{E0} = \bar{J}_{I0} = 4$. In Fig. 3, the parameters in the premotor network were: $\bar{J}_{EE} = 0.3$, $\bar{J}_{IE} = 6$, $\bar{J}_{EI} = -1.8$, $\bar{J}_{II} = -2.2$, $\bar{I}^E = 0.8$, $\bar{I}^I = 0.2$ and for the motor network: $\bar{J}_{E0} = \bar{J}_{I0} = 2$ and $\bar{J}_{EE} = 0.05$, $\bar{J}_{IE} = 0.75$, $\bar{J}_{EI} = -0.75$, $\bar{J}_{II} = -1$, $\bar{I}^E = 0.05$, $\bar{I}^I = 0.025$. $\tau_{eff} = 5$ ms. All synaptic time constants were 3 ms except for the premotor-to-motor pathway to the excitatory neurons in the motor network, which represents the slow NMDA synapses in the LMAN–RA pathways. The parameters used in the simulations depicted in Figs 4 and 6 were chosen such that the mean firing rates of the neurons in the premotor and motor networks were in agreement with previous experimental data as well as our own data in LMAN and RA. Given these parameters, the average firing rates of excitatory and inhibitory neurons in the premotor network were 14.7 and 46 Hz, consistent with our data and with previous reports for adult and juvenile finches[10]. The mean firing rates in the motor network are 40 Hz for the E cells and 100 Hz for the I cells, as reported for RA neurons[28]. The synaptic time constants of AMPA- and GABA$_A$-mediated synapses are all taken to be 3 ms. NMDA-mediated synapses in the premotor-to-motor pathway are modelled in a minimal manner, neglecting their voltage dependence, with very fast (instantaneous) rise and slow exponential decay with time constants of $\sim 100$ ms, in line with experiments[31].

We would like to stress here once more that the qualitative behaviour of the model is highly robust to changes in all its parameters (see Supplementary Note 1).

**Numerical integration.** The dynamics of the model circuit were numerically integrated using the Euler method supplemented with an interpolation estimate of the spike times[54]. In all simulations the integration time step was 0.1 ms. We verified the validity of the results by performing complementary simulations with smaller time steps.

**Autocovariance and crosscovariance of spike activities.** Neuronal spike trains were filtered with an exponential kernel (time constant = 5 ms). ACs and CCs of neuronal activities were estimated from the resulting smoothed signals. Population-averaged ACs and CCs were computed over all neurons in the corresponding population. The Pearson CC was defined as the crosscovariance normalized by the autocovariance at zero lag.

**Measure of synchrony and variability of the effectors.** We quantified the degree of synchrony in the activities of the premotor or motor network using the synchrony measure, $\chi(M)$, defined by[55,56]:

$$\chi^2 = \left[ \frac{\mathrm{Var}(m(M,t))}{\frac{1}{M}\sum \mathrm{Var}(v_i)} \right]$$

where the sum is over a population of $M$ neurons in the network, $v_i(t)$ is the instantaneous firing rate of neuron $i$ and $m(M,t) = \frac{1}{M}\sum v_i(t)$ is the instantaneous firing rate averaged over the population of M neurons. Here, $\mathrm{Var}(x(t))$ denotes the variance of the temporal fluctuations of $x(t)$ and $[x]$ denotes the average over a large number of realizations of the population S. For $1 \ll M \ll N$: $\chi^2(M) \simeq a(N) + \frac{b}{M}$, where $a$ and $b$ are numbers which depend on the network parameters. By definition, the network is in an asynchronous state if $a$ vanishes for sufficiently large $N$. In that case, pair-wise correlations are small, of the order of 1/$N$ and the population average firing rate is constant in time. In contrast, if $a$ converges to a non-zero value for large $N$, the network is in a synchronous state. To quantify the variability of the inputs to the effectors (receiving inputs from $M$ neurons in the motor network), $E_l(M, t)(l = 1, \ldots, D)$, we computed the coefficient of variation, $\mathrm{CV}_{\mathrm{eff}}$ such that:

$$\mathrm{CV}_{\mathrm{eff}}^2 = \left( \frac{1}{D}\sum_{l=1}^{D} \frac{\mathrm{S.d.}(E_l(M))}{\mathrm{Mean}(E_l(M))} \right)^2 = A + \frac{B}{M}.$$

If the motor network is in an asynchronous state, $A \sim \frac{1}{N}$, since $E_l(t)$ is linearly related to the population averaged activity in a functional group $l$.

**Data availability.** All relevant data and computer codes are available from the authors.

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

## Acknowledgements

We thank Shaul Druckmann, Robert Gütig, Carole Levenes, Gianluigi Mongillo, Richard Mooney, Israel Nelken, David Perkel, Nicholas Priebe, Frederic Theunissen and Tatyana Sharpee for comments on the manuscript. We thank Catherine Del-Negro and Karine Martel for sharing with us some of their behavioural data and the parents of the human infants for recordings their infants. Work conducted in the framework of the France Israel Laboratory of Neuroscience. Grants: ANR SONGLEARN, ANR. BALWM, Marie Curie IRG-NMVLRBG, France-Israel High Council for Science and technology, and LIA FILN.

## Author contributions

R.D. and A.L. performed the behavioural data analysis. S.P. provided the sparrows data. W.E.W. and A.L. performed the electrophysiological experiments. R.D., W.E.W. and A.L. performed the electrophysiological data analysis. R.D. and D.H. designed the theory and performed the simulations. R.D., A.L. and D.H. designed the study and wrote the manuscript.

## Additional information

**Competing interests:** The authors declare no competing financial interests.

