## [Peer Review File · Nature Communications]

Editorial Note: This manuscript has been previously reviewed at another journal that is not operating a transparent peer review scheme. This document only contains reviewer comments and rebuttal letters for versions considered at Nature Communications. Mentions of prior referee reports have been redacted.

PEER REVIEW FILE

Reviewers' Comments:

Reviewer #1 (Remarks to the Author):

I'm now happy with the paper. Only two comments.

1. Rosenbaum et al. (Nature Neurosci., 20:107-114, 2017) recently published a paper with virtually identical theory; that paper should be referenced.
2. The explanation of why there are large correlated fluctuations in the motor network is, I think, extremely hard to follow; the authors should make a much stronger effort to make it understandable. In particular, the arguments on page 6 make it seem like the input to the motor network is order(K), so that fluctuations scale as \sqrt{K} -- which is very large, and not possible to balance with nonspecific inhibition. Presumably, the strengths of the feedforward connections scale as $1/\sqrt{K}$, but that's hard to extract.

I'm guessing some of the explanation will have to go in Supplementary Material. If so, then when referring the reader to SM, the relevant section should be specified (SM is pretty long).

That's a general statement: all references to SM should have section information. That will make it much easier for those of us who care about the details.

Reviewer #2 (Remarks to the Author):

The authors have been extremely thorough in the reply to reviewers and have satisfactorily addressed all my previous concerns. The manuscript is technically novel, mathematically elegant and with interesting support from experimental data. I would be happy to see the paper published in its current form.

Typos:

spaces are often missing in the version of the text with highlighted manuscript changes, possibly due to some conversion issue (lines 59, 115, 116, 118, 123,153,169,193,204, 220, 225,284,300,317,495,550)

there's an extra . in line 220

Reviewer #3 (Remarks to the Author):

The main point of this manuscript is understanding how to generate behavioral variability--a key part of motor exploration. The authors show that it is not via the trivial consequences of neuronal variability, that in fact a particular circuit structure is needed. The authors show--via modeling and songbird physiology--that a circuit with topographic feedforward and recurrent connections can generate behavioral variability. Such a circuit eschews the problem that neuronal variability is typically averaged out. The authors then provide behavioral data and show that their statistics are also consistent with the model.

I'm a third reviewer for this manuscript and did not participate in the original review. Overall, I found that this study put forth a very interesting problem that is of interest to anyone studying development and behavioral variability. The authors proposed a solution, tested their hypotheses with a model and with physiology and behavioral data. I found these to be convincing overall.

My one concern is the following: For the across species analysis of vocal behavior, the authors used a single biomechanical model for sound generation. This model was constructed to mimic the zebra finch (Amador et al). I found the way they used this model to be very clever. However, contrary to the claims of Elemans et al. 2015 (at least in their title), it is well known that there are quite diverse mechanisms of vocal production even across songbird species (see the works of F. Goller, J.M. Wild and R. A. Suthers). Thus, I'm skeptical of the results in the present that report commonalities in the statistics of multiple songbirds and humans that seem to be all based on this single biomechanical model. Indeed, that model itself had to be modified from its original human instantiation (Titze, 1988) to account for zebra finch singing. Using the identical model for all species may spuriously generate the similar statistics.

The authors may do what they wish, but I'd be more comfortable if they only presented the zebra finch behavioral data.

Response to Reviewers' Comments

Text sections in italics are copied from comments of the reviewers .

Reviewer #1:

I'm now happy with the paper. Only two comments.

1. Rosenbaum et al. (Nature Neurosci., 20:107-114, 2017) recently published a paper with virtually identical theory; that paper should be referenced.

We reference this paper as follows at the end of the Discussion:

During the review process we became aware of a manuscript⁵⁶ which partially overlaps our work. It addresses the origin of spatially correlated activity in macaque V1. The mechanism proposed in this paper relies, like ours, on structured feedforward excitation and broad recurrent interactions.

2. The explanation of why there are large correlated fluctuations in the motor network is, I think, extremely hard to follow; the authors should make a much stronger effort to make it understandable. In particular, the arguments on page 6 make it seem like the input to the motor network is order(K), so that fluctuations scale as \sqrt{K} -- which is very large, and not possible to balance with nonspecific inhibition. Presumably, the strengths of the feedforward connections scale as $1/\sqrt{K}$, but that's hard to extract.

I'm guessing some of the explanation will have to go in Supplementary Material. If so, then when referring the reader to SM, the relevant section should be specified (SM is pretty long).

We have modified the text to better explain why there are large correlated fluctuations in the motor network.

“Neural activity in the motor network will be spatially correlated if the amplitude of the fluctuations in the structured component to the network is on the order of the neuronal threshold. This implies that the temporally averaged FF input must be on the order of \sqrt{K} . This will happen if the strength of the feedforward connections are on the order of $1/\sqrt{K}$. To prevent the neurons in the motor network to fire regularly at a very high rate, the inhibitory recurrent inputs in the motor network must compensate for most of this averaged FF input. Such a compensation occurs naturally if the motor network is strongly recurrent and operates in the ‘balanced excitation-inhibition’ regime¹³ (see Supplementary Note.1 for more details on the mechanism).”

That's a general statement: all references to SM should have section information. That will make it much easier for those of us who care about the details.

Done

Reviewer #2:

The authors have been extremely thorough in the reply to reviewers and have satisfactorily addressed all my previous concerns. The manuscript is technically novel, mathematically elegant and with interesting support from experimental data. I would be happy to see the paper published in its current form.

Typos:

spaces are often missing in the version of the text with highlighted manuscript changes, possibly due to some conversion issue (lines 59, 115, 116, 118, 123,153,169,193,204, 220, 225,284,300,317,495,550)

there's an extra . in line 220

We have corrected these typos.

Reviewer #3:

The main point of this manuscript is understanding how to generate behavioral variability--a key part of motor exploration. The authors show that it is not via the trivial consequences of neuronal variability, that in fact a particular circuit structure is needed. The authors show--via modeling and songbird physiology--that a circuit with topographic feedforward and recurrent connections can generate behavioral variability. Such a circuit eschews the problem that neuronal variability is typically averaged out. The authors then provide behavioral data and show that their statistics are also consistent with the model.

I'm a third reviewer for this manuscript and did not participate in the original review. Overall, I found that this study put forth a very interesting problem that is of interest to anyone studying development and behavioral variability. The authors proposed a solution, tested their hypotheses with a model and with physiology and behavioral data. I found these to be convincing overall.

My one concern is the following: For the across species analysis of vocal behavior, the authors used a single biomechanical model for sound generation. This model was constructed to mimic the zebra finch (Amador et al). I found the way they used this model to be very clever. However, contrary to the claims of Elemans et al. 2015 (at least in their title), it is well known that there are quite diverse mechanisms of vocal production even across songbird species (see the works of F. Goller, J.M. Wild and R. A. Suthers). Thus, I'm skeptical of the results in the present that report commonalities in the statistics of multiple songbirds and humans that seem to be all based on this single biomechanical model. Indeed, that model itself had to be modified from its original human instantiation (Titze, 1988) to account for zebra finch singing. Using the identical model for all species may spuriously generate the similar statistics

The authors may do what they wish, but I'd be more comfortable if they only presented the zebra finch behavioral data.

The biomechanical model we use for sound generation was specifically designed to mimic zebrafish. However, we argue in the paper as well as in the Supplementary Material (Supplementary Figure 8) that the features in the ACE and the distribution of the gesture durations (but clearly not the fine-structure within gestures, which can crucially depend on the mechanical organ of the species) reflect universal statistical properties of the circuit dynamics and are, to a large extent, insensitive to the circuit parameters and to the details of the transformation from the input to the effectors to the vocal behavior. The similarities in the statistics in the behavioral data we recorded in human babies and in songbirds are in line with our theoretical arguments. This is the reason we have kept the presentation of our human behavioral data in the final version of the paper.